# Endocrine Regulation of Maturation and Sex Change in Groupers

**DOI:** 10.3390/cells11050825

**Published:** 2022-02-27

**Authors:** Kiyoshi Soyano, Takafumi Amagai, Tomofumi Yamaguchi, Yuji Mushirobira, Wen-Gang Xu, Nhan Thành Phạm, Ryosuke Murata

**Affiliations:** 1Institute for East China Sea Research, Organization for Marine Sciences and Technology, Nagasaki University, 1551-7 Taira-machi, Nagasaki 851-2213, Japan; tamagai@nagasaki-u.ac.jp (T.A.); y.mushirobira@nagasaki-u.ac.jp (Y.M.); thanhnhanctu@gmail.com (N.T.P.); murata-r@nagasaki-u.ac.jp (R.M.); 2Graduate School of Fisheries and Environmental Sciences, Nagasaki University, 1-14 Bunkyo-machi, Nagasaki 852-8521, Japan; tynd@affrc.go.jp; 3Fisheries Technology Institute, Japan Fisheries Research and Education Agency, 148 Fukaiota, Ishigaki, Okinawa 907-0451, Japan; 4School of Ocean, Yantai University, 30 Qingquan RD, Laishan District, Yantai 264005, China; xugang@ytu.edu.cn; 5College of Aquaculture and Fisheries, Can Tho University, 3/2 Street, Xuan Khanh Ward, Ninh Kieu District, Can Tho City 900000, Vietnam

**Keywords:** groupers, vitellogenesis, final oocyte maturation, spawning, sex change

## Abstract

Groupers are widely distributed in tropical and subtropical areas worldwide, are key species to coastal ecosystems, and valuable fishery targets. To facilitate artificial seed production technology for grouper aquaculture, the mechanisms of reproduction and gonad development are being elucidated for these important species. In addition, since groupers are sexually dimorphic fish with female-first maturity (protogynous hermaphrodite fish), research is being conducted to clarify the ecological mechanism of sex change and their reproductive physiology, focusing on the endocrine system. In recent years, research on groupers has also been conducted to understand changes in the coastal environment caused by ocean warming and man-made chemicals. However, due to difficulties associated with conducting research using wild populations for breeding experiments, knowledge of the physiology and ecology of these fish is lacking, especially their reproductive physiology. In this review, we present information on the reproductive physiology and endocrinology of groupers obtained to date, together with the characteristics of their life history.

## 1. Introduction

Groupers, a species of fish belonging to the family Serranidae, are protogynous hermaphrodites, meaning the first spawning is by females, who later transform into males [1]. These species are widely distributed in subtropical and tropical areas worldwide, obtaining high market price due to their popularity in Japanese, Chinese, and Western cuisine. Considered “ultra-luxury fish”, groupers are among the most commercially valuable species in Asia. Moreover, groupers are considered key species in coastal ecosystems, and their decline due to fishing pressure has a significant impact on the ecosystem. Therefore, overfishing to meet market demand is a concern. As a countermeasure for their preservation and sustainable use, artificial seed production of groupers is actively carried out in various countries around the world, especially in East Asian countries [2,3]. In Japan and Asian countries, the red spotted grouper (*Epinephelus akkara*), seven-band grouper (*E. septemfasciatus*), longtooth grouper (*E. bruneus*), Malabar grouper (*E. malabaricus*), white-streaked grouper (*E. ongus*), areolate grouper (*E. areolatus*), and leopard coral grouper (*Plectropomus leopardus*) are successfully mass produced using artificial seed [2,3]. Studies on reproductive physiology based on seed production, induction of maturation and spawning, and sex change have been conducted on several economically important and easily accessible grouper species, such as the seven-band grouper [4], longtooth grouper [5], red spotted grouper [6], honeycomb grouper (*E. merra*) [7,8], and blacktip grouper (*E. fasciatus*) [9]. However, these studies have not sufficiently elucidated grouper reproduction.

The main feature of grouper reproduction is the occurrence protogynous hermaphrodism, with all individuals first differentiating into females in the early stages of development [10,11]. Additionally, groupers are known to gather at spawning grounds for spawning [12]. These phenomena and characteristics are important for understanding the maturation mechanisms of groupers.

Unfortunately, knowledge of grouper reproduction is restricted by difficulties associated with capturing these species and observing them in natural waters. However, research has been actively conducted in recent years, and interesting findings are being reported. This family of fish thus provides an interesting model for fish reproductive physiology. In this review, we present current research on grouper reproduction, describing the following points from the perspectives of reproductive physiology and endocrinology: (1) first maturity (puberty), (2) final maturation and spawning, and (3) sex change and male maturation. The maturation process of the grouper described in this review and the major endocrine factors affecting these maturation phenomena are shown in Figure 1.

## 2. First Maturity (Puberty)

### 2.1. Mechanism of Maturation in Grouper

The process by which an animal acquires fertility for the first time is called first maturity or puberty [13,14]. Understanding the age of puberty for each fish species is extremely important for resource management in aquaculture. In particular, clarifying the age and size at which the target species reaches puberty is necessary when cultivating parent fish for artificial seed production. However, the conditions for initiating first maturity differ depending on the fish species. Moreover, the onset of puberty is controlled by activation of the brain–pituitary–gonad (BPG) axis, which can be artificially controlled through hormonal and environmental manipulation of the reproductive endocrine system [15,16,17].

It remains unclear whether age or body size is important for the initial maturity of groupers, since the endocrine mechanisms involved have yet to be elucidated. It is known among fishermen and fish marketers that the body size of mature honeycomb grouper and longtooth grouper differed depending on the habitat. However, no studies have been conducted to support this issue, and scientific evidence is lacking. Nevertheless, rapid growth is positively correlated with early puberty in fish [16]. Moreover, puberty onset is controlled by the activation of the BPG axis, and a range of internal and external factors are hypothesized to stimulate and/or modulate this activation, such as growth, adiposity, feed intake, photoperiod, temperature, and social factors.

Does the endocrine system change with size or age in groupers? In fact, there is no clear evidence that age affects the induction of endocrine signals at the onset of first maturity. However, size reportedly affected hormone secretion in the pituitary gland and reproductive endocrine system during puberty in cultured longtooth grouper [18]. Although culture conditions differ from those in a wild environment, this information provides clues that contribute to our understanding of the endocrine system during puberty in groupers.

The key hormone to control reproductive phenomena is gonadotropins (GtHs), which is secreted by the pituitary gland. GtHs include two different forms: follicle-stimulating hormone (FSH) and luteinizing hormone (LH). The functions of FSH and LH in oocyte development, maturation, and sex change will be discussed later, but they also play an important role in puberty. Immunohistochemistry in the pituitary gland of the red spotted grouper using antibodies against FSH and LH showed that the positive reaction appeared in the cells of central, ventral, or the marginal part of the proximal pars distalis area before ovarian differentiation and continued to appear thereafter [19]. However, there is no detailed information on the expression of these hormones before puberty; unfortunately, the role of both in puberty is not clear. Cellular and molecular studies on these hormones are essential to understand the endocrine regulation during puberty in groupers.

### 2.2. Vitellogenesis

The first step towards spawning in fish is yolk formation, called vitellogenesis. Vitellogenesis is the process of accumulating yolk protein into oocytes, which occurs under the control of GtHs. The role of GtHs in this process is to induce the synthesis of estradiol-17β (E2) in the ovarian follicle layer, which triggers the production of vitellogenin (Vtg) in the liver and causes oocyte growth [20]. Both FSH and LH, different forms of GtHs, are composed of a common α-subunit and hormone-specific β-subunit [21,22]. The GtH gene has already been cloned and sequenced in several fish species, enabling the deduction of the amino acid sequence of the GtH gene in groupers [23]. Recombinants and antibodies have also been created using this information [24]. FSH and LH, produced in independent adenohypophyseal cells, are transported to the gonads through blood vessels [25], while gametogenesis is induced via steroidogenesis in the ovaries. The patterns of FSH and LH mRNA expression and protein synthesis differ during the reproductive cycle [26,27,28]. For example, FSH is known to act on yolk formation while LH induces the final maturation in salmonids [29]. However, in many marine fish, the functions of these two GtHs are not clearly separated. In groupers, LH is thought to be responsible for yolk formation (see below for details) [30]. In teleost fish, GtH generally synthesizes E2 from cholesterol in the two cell layers surrounding the oocyte (theca and granulosa cell layers) [29]. This is also the case in groupers, where E2 acts on the liver to synthesize Vtg. Vtg is taken up by the oocyte via Vtg receptors [31,32]. Vtg is a precursor of yolk protein and is resolved into four proteins, including lipovitellin, phosvitin, β’-component, and c-terminal protein, immediately after being taken up by the oocyte. These proteins are generally called yolk proteins. Although changes in GtH, sex hormones, and Vtg levels during maturation have been well studied in many fish, there is little information about these changes in groupers. In the red spotted grouper, red grouper (*E. morio*), and leopard coral grouper, plasma E2 levels increased during vitellogenesis a few months before spawning [33,34,35]. Measurement of plasma LH levels using the LH ELISA system established for groupers confirmed that LH levels were low in individuals with immature oocytes but high in those with developed oocytes [24]. Thus, the synthesis of LH was correlated with oocyte development. Vtg, under the control of these hormones, was detected in pre-mature females using a non-invasive method based on a vitellogenin immunoassay for skin mucus [36]. There are no reports of annual changes in plasma Vtg levels for groupers, but results with other fish support that gonadal development is correlated with these hormones and Vtg profiles [31]. Therefore, we hypothesized that Vtg levels would also increase in groupers during the spawning season. Accordingly, the only measurement of plasma Vtg levels in groupers was conducted on the basis of sex discrimination at maturity, revealing that the Vtg was detected in mature females, but not in males [12]. Thus, Vtg, the expression of which increases during maturation, was taken up by the oocyte and accumulated as a yolk protein. Its accumulation in groupers occurred over a period of approximately one to two months before spawning [4,7,37].

The Vtg-derived yolk protein is taken up by the oocyte and broken down for use as nutrition for hatchling larval fish. However, the role of yolk protein varies among fish species. For example, yolk protein is also used for buoyancy in pelagic eggs [38,39,40]. The different roles of yolk proteins are related to the different Vtg subtypes, which have been reported in many fish species and play different roles. Vtg sequence and amino acid analyses were recently conducted for the giant grouper (*E. lanceolatus*), revealing similar linear yolk protein domains (lipovitellin heavy chain, phospvitin, lipovitellin light chain, and β-component) to those of Vtg from the sailfin molly (*Poecilia latipinna*) and mangrove killifish (*Kryptolebias marmoratus*) [41]. Role-differentiated Vtg subtypes found in other marine fish with pelagic eggs have not yet been identified in groupers, although three different Vtg subtypes have been identified in fish belonging to the same Perciformes order as groupers [31]. Therefore, it is possible that different Vtg subtypes are present in groupers. Unfortunately, this information is less available for groupers than other fish species, and gene and amino acid sequence data for different types of Vtg need to be collected for many more grouper species.

What kinds of environmental stimuli induces hormone synthesis to control the development and maturation of gonads? Below, we introduce the relationship between environmental factors and maturity in groupers living in Japanese coastal waters as an example.

Northern-type groupers, such as the seven-band grouper, longtooth grouper, and red spotted grouper, prefer relatively cold waters, while southern types, such as the blacktip grouper, leopard coral grouper, and Malabar grouper, inhabit the warmer coastal waters of Japan. Histological observation of the development of the seven-band grouper ovary revealed that oocyte development began after February when the water temperature began to rise, and yolk accumulation progressed rapidly from April to May [4]. Controlling the development and maturation of seven-band grouper gonads was possible by manipulating both photoperiod and water temperature, inducing maturation and spawning at a different time from the natural spawning season [42]. These results indicate that water temperature and photoperiod regulate the maturation process. In addition, gonad development of the red spotted grouper was induced by raising the rearing water temperature from 18 °C to 23 °C and extending the photoperiod from 12 L: 12D to 14 L: 10D [43]. This result further supports that water temperature and photoperiod play important roles in inducing maturation. In the case of northern-type groupers, such as the seven-band grouper and red spotted grouper, is water temperature or photoperiod the effective factor? Even under the same water temperature conditions as in early summer, gonad development is not observed in autumn when day length is shorter, suggesting that gonadal development and spawning in northern-type groupers are controlled by both factors. Interestingly, in the experiment by Lee [43], the expression of FSH and LH subunit genes (*fshb* and *lhb*) in the pituitary gland was significantly increased in groupers reared in warmer water with a longer photoperiod. This is a strong indication that water temperature and photoperiod activate the HPG axis.

In contrast, it is possible to induce maturation and collect fertilized eggs year-round in southern-type groupers, such as the blacktip grouper, by controlling the water temperature [44]. In our laboratory, blacktip grouper spawned repeatedly at suitable water temperatures under long-day conditions in June until short-day conditions in October. In other words, maturation and spawning of southern-type groupers are not affected by photoperiod, but depend on water temperature. However, maturation was successfully induced in honeycomb grouper (also southern type) during the non-spawning period by manipulating both photoperiod and water temperature along with hormone administration [45]. In this experiment, the combined effects of photoperiod (long/short) and water temperature (low/high) were examined on maturation in non-spawning individuals administered gonadotropin-releasing hormone (GnRH), revealing that longer days and higher water temperatures were necessary for the induction of maturation. Therefore, it cannot be unequivocally stated that day length has no influence on the reproductive endocrine system.

## 3. Final Maturation and Spawning

Final oocyte maturation (FOM) refers to a series of physiological changes in oocytes culminating in ovulation, such as nucleus (germinal vesicle) migration to the animal pole, germinal vesicle breakdown (GVBD), and aggregation of yolk granules and oil droplets with water absorption. These phenomena are promoted by the final maturation-inducing hormone (MIH), which is induced by a surge in LH secreted from the pituitary gland at this time [20,46]. Prior to this phenomenon, oocytes that have completed yolk accumulation must acquire sensitivity to MIH, which induces FOM. Acquisition of oocyte maturational competence (OMC), which is induced by LH, is similar to FOM [47,48]. OMC in females is thus the acquisition of MIH sensitivity in oocytes.

Some grouper species do not spawn even in a suitable spawning environment, likely due to the inhibition of gonadal development and/or spawning behavior. Groupers have a unique swimming behavior when spawning approaches, in which male fish chase mature female fish and they swim toward the surface in pairs [49,50]. However, spawning performance declines if the distance required for swimming is insufficient, or if there is insufficient space for spawning behavior [50]. Nevertheless, it is not only the physical spawning environment that controls grouper spawning. Cultivation of only female Malabar grouper inhibited the progress of oocyte maturation and ovulation (according to a report from a Malabar grouper farm). This result indicates that the presence of males plays an important role in the induction of oocyte maturation and ovulation in female groupers. Therefore, we hypothesized that male and female groupers might communicate, possibly via pheromones, prior to final maturation and spawning. Conducting rearing experiments using a honeycomb grouper with completely accumulated yolk protein, males and females were divided into separate tanks before the spawning season, and the rearing water from male-only or empty (control) upper aquariums flowed into lower aquariums containing fully vitellogenic females during the spawning phase. To avoid visual cues, the structure of the rearing system was designed so the females in the lower aquarium could not visually recognize the males in the upper aquarium (Figure 2). The results indicated that final maturation progressed and ovulation occurred in mature females exposed to the rearing water of mature males, but not that of the control tank [51]. These results demonstrate that the execution of FOM and ovulation in honeycomb grouper females requires pheromones from mature males. Pheromones are important signaling substances in many organisms. Peter and Yu [52] suggest that pheromones can stimulate physiological changes necessary for reproduction. Indeed, male pheromones have reportedly induced ovulation in females of some freshwater species [53,54,55,56]. There are no reported studies of pheromones inducing ovulation and spawning in marine fish, although the role of pheromones in triggering endocrine mechanisms associated with final maturation has been reported in rainbow trout (*Oncorhynchus mykiss*) [57] and sea lamprey (*Petromyzon marinus*) [58]. Pheromones have also been identified in these fish species, including sulfated sex steroids in rainbow trout [57] and bile acids in sea lamprey [58]. In addition, a sex steroid-inducing pheromone effect has been reported in the urine of the tiger grouper (*E. fuscoquttatus)*, suggesting steroid compounds as pheromone candidates [59]. Thus, pheromone substances seem to vary between fish species. The removal of males from the environment of spawning females negatively affected spawning frequency, ovarian fecundity, and sex hormone levels on the reproductive axis of gilt-head bream (*Sparus aurata*) [60]. However, this study did not prove the role of pheromones. Thus, information on pheromones and phenomena related to the induction of maturation remains fragmented. In this context, our study with honeycomb grouper strongly indicates that pheromones released by males are important for FOM and induction of spawning, although we have not yet identified the pheromone substance. A suitable model is needed to study fish reproductive pheromones, which could be provided by groupers. It may be possible to identify pheromones and elucidate their role in fish reproduction by using groupers as model fish.

When is OMC acquired? FOM and ovulation are known to be induced by hormone administration, such as human chorionic gonadotropin (HCG) and luteinizing hormone-releasing hormone (LHRH), for artificial seed production in groupers [61]. However, not every fish with vitellogenic oocytes will respond to hormones and initiate the process toward FOM. The relationship between hormone response and oocyte diameter has been examined in many fish species, demonstrating species-specific variations. Although no scientific evidence is available, oocyte diameter is presumably related to the amount of yolk protein required for larval growth, and there must be a mechanism for transitioning to the next step, FOM, after sufficient yolk protein accumulation. Hormone-responsive minimum oocyte diameter has also been examined in some groupers. The sufficient oocyte diameter to enable hormone response is 450 µm in the seven-band grouper [54,61]. FOM and ovulation progress when HCG or LHRH is administered to oocytes with larger diameters, but no hormone response is observed at smaller oocyte diameters. In other words, OMC has not been acquired. Similar results have been reported in many fish species, suggesting that LH promotes the acquisition of OMC and secretion of MIH to advance the FOM process. However, many factors remain unclear for groupers, including whether pheromones from males are involved in the OMC and FOM processes.

Final oocyte maturation (FOM) of female reared in the tank with male rearing water is induced, resulting the rearing water of mature males contains pheromones necessary for the induction of FOM.

Control group: the females were reared alone with natural sea water without male; Male water group: the females were reared with mature male rearing water.

## 4. Sex Change and Males

### 4.1. Process of Sex Change

As mentioned above, groupers are hermaphrodites [1]. In the early stages of development, all individuals first differentiate into females and experience functional maturity [62,63,64,65,66], changing to functional males thereafter. Sex change is common in tropical and subtropical fish species, with protogynous hermaphrodite types being reported in wrasses [67,68] and gobies [69]. Moreover, there are several different forms of sex change. Protandrous hermaphrodism occurs in anemone fish [70,71], while bidirectional sex reversal occurs in the Okinawa rubble goby (*Trimma okinawae*) [72]. Sex change is often caused by changes in the social environment and structure [73,74], such as in the sex change of the halfmoon grouper (*E. rivulatus*) [75] and orange spotted grouper (*E. coioides*) [66,76]. A change in sociality caused by the collapse of territory due to the death or escape of a harem male is thought to be a common factor that induces sex change in many sexually transitional fish (Figure 3). These social factors act as stressors to induce sex change [74] and, as a result, may regulate sex hormone secretion by altering GnRH and GtH secretion and receptor expression. However, body size, not sociality, has been shown to act as a possible trigger for sex change in groupers. In the greasy grouper (*E. tauvina*), sex change occurs in a size-dependent manner [77]. Moreover, sex change in males was only observed in individuals over 20 cm in body length in the honeycomb grouper [63,64]. These results support the possibility that sex change in groupers is dependent on size. Nevertheless, since size at sex change also varies depending on the collection site, both sociality and body size should be considered as possible factors for sex change. Unfortunately, it is not clear how such factors are replaced by endogenous information to trigger the reproductive endocrine system. Thus, although there are still many uncertainties regarding the mechanisms that cause sex change, it is certain that sex steroids are a direct factor involved in sex change mechanisms. Below, we focus on the functions of GtH and sex hormones in sex change, using data from experiments with the honeycomb grouper (Figure 3).

Among sex steroids, androgens are presumed to directly induce sex change in the honeycomb grouper. The plasma levels of androgens (such as 11-ketotestosterone, 11-KT) and estrogens (such as E2) in the honeycomb grouper during sex change indicated that females and early-stage sex change individuals had low levels of 11-KT, while those in late-stage sex change and complete males had high levels [64,65,78]. Furthermore, immunohistochemical analysis of the expression of steroid-metabolizing enzymes (Cyp11a and Cyp11b) involved in their production in the gonads indicated that these positive reactions increased as sex change progressed, and the nucleus size in steroid-producing cells also increased [79]. The steroid-producing cells are already distributed in the ovarian tunica even at female stage, and then appear in the interstitial area of the gonads as Leydig cells, which is a major androgen-producing cell in the testes of vertebrates, with sex change progression [78]. These results indicate that 11-KT is the main factor inducing sex change in the honeycomb grouper. Recently, the steroid-producing cells distributed in the tunica gonads are commonly confirmed in almost all groupers, which are named TIS cells (testicular inducing steroidogenic cells), suggesting that 11-KT is the common main factor inducing sex change in groupers [80]. Interestingly, in wrasses, plasma E2 levels decreased sharply in individuals in the early stages of sex change [67]. Moreover, the increase in 11-KT levels occurring later than the decrease in E2 levels suggests that the latter plays an important role in the initiation of sex change. However, E2 levels do not decrease during the process of sex change in the honeycomb grouper, and E2 is always present at the same concentration during the sex change phase. Therefore, the effect of E2 on the initiation and progression of sex change is considered to be small [64].

The expression of GtH genes (*fshb* and *lhb*), which mediate the synthesis of sex steroids, was measured throughout the sex change process in the honeycomb grouper, revealing no difference in the expression of *lhb* in immature females before sex change from that of immature males that had undergone complete sex change [30]. In contrast, the expression of *fshb* in males, regardless of the degree of maturity, was higher than that in females. However, the expression of *fshb* was significantly higher in males immediately after sex change initiation. These results indicate that FSH plays an important role in male sex change groupers [30]. Furthermore, this experiment revealed some other interesting findings. When bovine FSH was administered to an immature female honeycomb grouper, active spermatogenesis was observed in the ovaries. This treatment increased 11-KT plasma levels, without a corresponding change in E2 levels, supporting the kinetics of plasma steroids in individuals undergoing a sex change. The expression of the FSH receptor gene in the gonads also increased with sex change [81]. These results indicate that FSH induces sex changes in males by stimulating the production of 11-KT. In addition, the results of this study support that FSH is the male form of GtH, as mentioned above.

Although androgens are indisputably involved in sex change in groupers, sex change can also be induced by an aromatase inhibitor (AI, Fadrozole), which inhibits Cyp19A1, the promotor of androgen-to-estrogen conversion. In anemone fish, which are protandrous hermaphrodites, administration of AI to mature females decreased plasma E2 levels and induced a sex change from female to male [71]. Similarly, AI administered to a female honeycomb grouper resulted in reduced plasma E2 levels and transformation into males [64]. Conversely, combined administration of AI and E2 did not induce a sex change from female to male. This is likely because the administered E2 compensated for the endogenous E2 suppressed by AI [78]. In contrast, administration of 17α-methyltestosterone (MT), a synthetic androgen, resulted in female-to-male conversion, further supporting that androgens effectively induce a sex change in groupers [82,83]. In the study of AI-induced sex change in the males mentioned above, the results may be due to the inhibition of E2 synthesis by AI, which not only decreases estrogen dominance, but also increases the effect of its precursor, androgen. These findings suggest that sex change may involve both a decrease in estrogen levels and an increase in androgen levels, or a balance between the two [84].

### 4.2. Male Maturation

Post sex change, males undergo spermatogenesis under the control of androgens. This endocrine mechanism is essentially the same as that of male gametogenesis in common fish [85]. However, in the case of groupers, steroid-producing cells with an enzyme (Cyp11b) involved in the production of androgens are already present in the ovary prior to sex change [80,86]. These cells are thought to be triggered to activate sex changes. Alternatively, MT can be artificially administered to induce androgenic functional spermatogenesis [87]. When MT was administered to immature female Malabar grouper, testes with active spermatogenesis appeared after approximately six months [88], indicating that androgens promote spermatogenesis. Similarly, spermatogenesis was induced by 11-KT treatment in the female honeycomb grouper. Moreover, when female-to-male sex change individuals were reared together with mature females, fertilized eggs were obtained, indicating that males induced by androgen treatment had functional sperm and a normal reproductive behavior [89].

Interestingly, when MT was administered to an immature female Malaber grouper to induce sex change, they reverted to immature females when MT treatment was discontinued [90]. Similar results have been reported with other grouper species. The MT treatment of an orange spotted grouper in the sexual differentiation phase caused differentiation into males, but they reverted to females when MT treatment was terminated [91]. In other words, immature individuals prior to sex change do not have sufficient endogenous androgen-producing functions in their gonads, becoming male only due to the temporary action of exogenous androgens. Since temporary males have not been observed in wild groupers, establishment as a functional male may be related to the maturity experience as a female.

## 5. Conclusions and Perspectives

Groupers are an economically valuable species that occupy an important position in coastal ecosystems. With increasing fishing pressure in recent years, groupers have become a threatened species, with rapidly declining populations. Resource management practices are thus necessary to protect grouper populations, such as restricting fishing at spawning grounds due to the tendency of groupers to aggregate at suitable spawning grounds. However, the implementation of such measures necessitates fully understanding the physiological and ecological characteristics of grouper reproduction in the natural environment. Particularly noteworthy features include migration from habitat to spawning grounds, communication between males and females at spawning grounds, the characteristics of spawning behavior, changes in reproductive physiology caused by such communication, and environmental factors that cause such behavior (Figure 4). Without understanding these factors, it is impossible to formulate measures to preserve grouper populations. Groupers are also excellent models for understanding the reproductive strategies of fish and their associated reproductive mechanisms. There are many interesting phenomena associated with grouper reproduction, such as sex change, harem formation, gonad development, and spawning synchronized with the lunar cycle. The reproductive phenomena in groupers are regulated by endocrine factors associated with those reproductive strategy, which are not found in other fish species. One of the most notable characteristics of groupers is that they are protogynous hermaphrodites. Sex determination in this species is controlled by a sociality (harem) centered on the male. In addition to sex determination, the presence of males and females in groupers has a significant impact on their reproductive phenomena. The ovulation and spawning of females require pheromonal stimulation from males, which activates the endocrine system for final maturation. The ovaries of females have interesting histological and cytological features, including the presence of cells that already synthesize androgen. Another interesting feature is that sex change is endocrinologically triggered by increased secretion of FSH, the male type of GtH in groupers, in hormone-producing cells of the pituitary gland, resulting in increased 11-KT. However, there are still many unknowns, including the process of puberty and first maturing as a female. Elucidating the exogenous and endogenous factors controlling reproduction and the hormone secretion in the brain–pituitary–gonadal axis in response to these factors at the cellular and molecular levels is important for understanding the reproductive phenomena in groupers. These findings will also lead to improved understanding of the biological characteristics of groupers, while facilitating the conservation of wild populations and development of improved aquaculture techniques. However, much research is still needed to elucidate the eco-physiology of grouper reproduction. We hope that this review will encourage fish physiology and endocrinology researchers to participate in the study of grouper reproduction.

Relationships among the migration from habitat to spawning grounds, maturation, physiological changes, and endocrine changes in the grouper.

The southern type groupers, including the honeycomb grouper, are lunar-related spawners. They migrate to suitable spawning grounds just after the full moon for final oocyte maturation and spawning.

## Figures and Tables

**Figure 1 cells-11-00825-f001:**
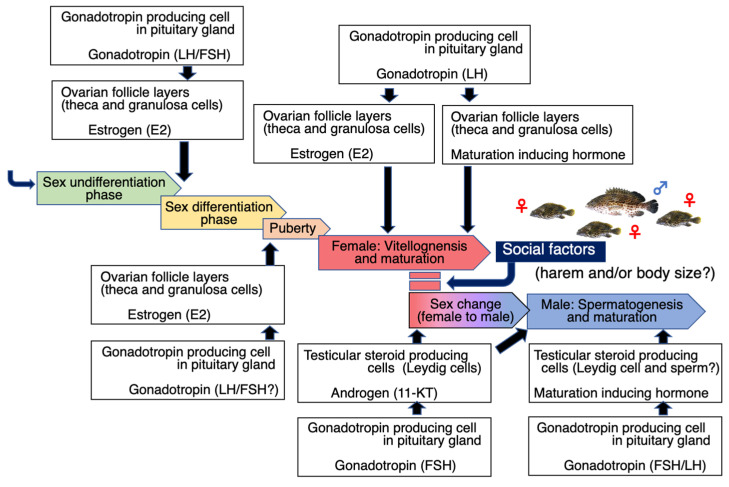
Gonadal development in groupers and the endocrine factors affecting these maturation phenomena.

**Figure 2 cells-11-00825-f002:**
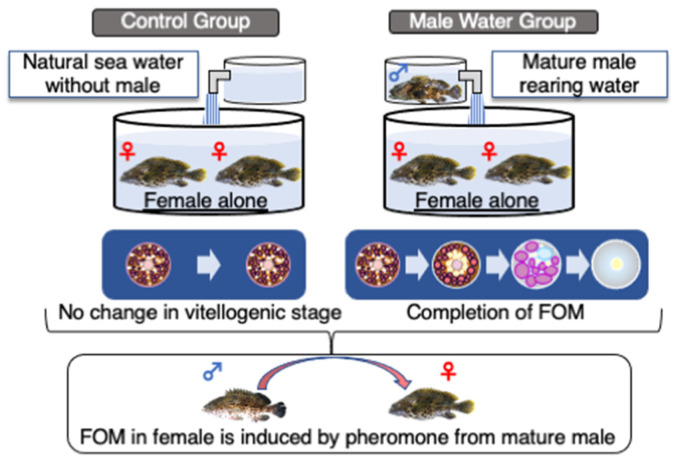
Induction of final maturation in females by pheromones released from mature males.

**Figure 3 cells-11-00825-f003:**
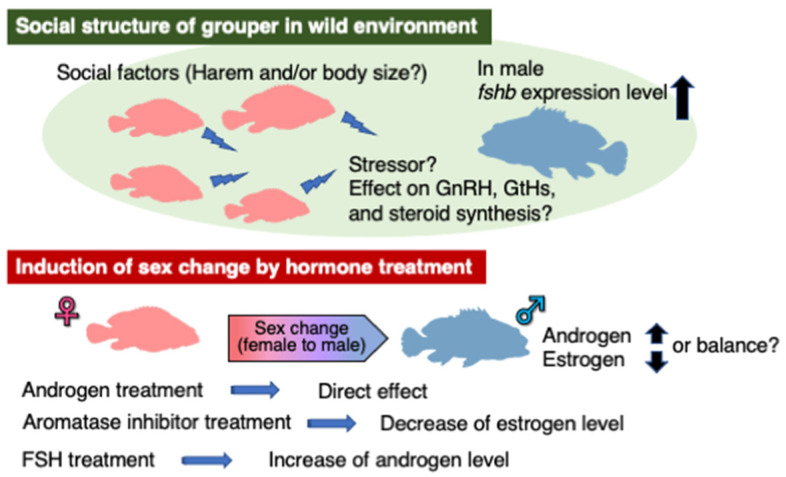
Factors involved in the induction of sex change in groupers.

**Figure 4 cells-11-00825-f004:**
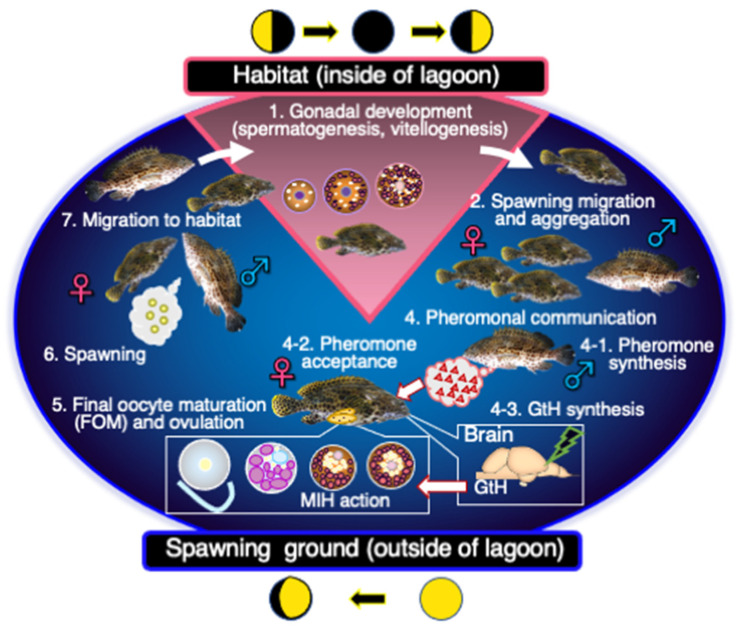
Maturation mechanisms of the honeycomb groupers in Okinawa, Japan.

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
