# Peer review of "Endocrine Regulation of Maturation and Sex Change in Groupers"

_cells, 2022, doi:10.3390/cells11050825_

Round 1
Reviewer 1 Report
The review, entitled "Endocrine regulation of maturation and sex change in groupers", clearly analyzes the reproductive physiological mechanisms and endocrinology of groupers.
The introduction is supported by bibliographic studies consistent with the topic of study.
The different sections are consistent with the content of the review and supported by bibliographic references.
the research is interesting to better understand the reproductive physiology and endocrinology of this type of animal.
The study is original and more in-depth than the literature, making the topic of study clearer and more comprehensive.
The text is therefore usable and well written and the conclusions are consistent with the topic studied highlighting also future perspectives in this regard.
Author Response
Thank you for your evaluation. I have added some information based on the comments of other reviewers.
Reviewer 2 Report
The MS is interesting and well written. I have two comments:
First, I think that unpublished results should be eliminated from the MS, as it does not have passed by a peer-review revision. For example, line 86; lines 212-213; line 224-225. Line 358.
Second, the conclusions and perspectives part seems to general; I recommend to elaborate an abstract about the main findings in the puberty, sex development and sex change in groupers
Author Response
Thank you for your useful and appropriate comments.
Point 1: First, I think that unpublished results should be eliminated from the MS, as it does not have passed by a peer-review revision. For example, line 86; lines 212-213; line 224-225. Line 358
Response 1: We have changed the description of the unpublished information as follows.
L86: We have removed the text with your suggestion. We have also revised the text before and after it.
L212-213:Information from the field of aquaculture is difficult to be published in a paper. However, this information is important as a starting point for the design of the study, so it was retained. (Revised MS, L315-317)
L224-225:Our paper has been accepted and we have added it. (Revised MS, L328)
L358: We have removed it, as you suggested.
Point 2: Second, the conclusions and perspectives part seems to general; I recommend to elaborate an abstract about the main findings in the puberty, sex development and sex change in groupers
Response 2: We have accepted your suggestion and added a note about a characteristic phenomenon in groupers in the Conclusion and Perspectives part. It is indicated by a yellow marker. (Revised MS, L590-606)
Reviewer 3 Report
Abstract: No comment
Introduction: No comment, but I have question, whether pollution of water changes any how groupers lifestyle.
In figure 1 what are the social factors? If you can mention in the figure.
First maturity: In line 90, what is BPG axis?
In 2.2, line 133, where sentence ends, it will be better to have a reference, since we use the term correlation. There are statistical term so whenever used, may need a reference.
Final maturation and spawning: In line 215, “possibly via pheromones” without any reference. Later pheromone occurrence mentioned with references, line 233, 234. If you can mention this references earlier that will be better.
Sex change and males: no comments
Conclusions and perspective: no comments
Author Response
Thank you for your useful and appropriate comments.
Point 1: Introduction: No comment, but I have question, whether pollution of water changes any how groupers lifestyle.
Response 1: As for the question of whether water pollution will affect the lifecycle of groupers, it is difficult to imagine any change in their lifecycle since they live in areas that are relatively non polluted. However, because of grouper has pheromonal communication during a process of reproductive phenomena, it is possible that heavy pollution could affect their reproductive success. Also, as a sex change fish, it is possible that environmental estrogens and other factors could affect its sex ratio. The possibility of changes in lifecycle as a result cannot be dismissed.
Point 2: In figure 1 what are the social factors? If you can mention in the figure.
Response 2: We have added social factors to Figure 1.
Point 3: First maturity: In line 90, what is BPG axis?
Response 3: We have a description of the BPG axis in line 77 (Revised MS, L84). Sometimes the hypothalamic-pituitary-gonadal axis (HPG axis) is used, but we use the term brain-pituitary-gonadal axis (BPG axis) because the input information is not controlled only by the hypothalamus.
Point 4: In 2.2, line 133, where sentence ends, it will be better to have a reference, since we use the term correlation. There are statistical term so whenever used, may need a reference.
Response 4: We have added a reference with your suggestion. (Revised MS, L219)
Point 5: Final maturation and spawning: In line 215, “possibly via pheromones” without any reference. Later pheromone occurrence mentioned with references, line 233, 234. If you can mention this references earlier that will be better.
Response 5: This text is an explanation of our hypothesis for the following experiment. Therefore, no references have been added here. However, a paper describing the next study has been accepted, and we have added it to the text that follows.